# Cardiomyopathy Associated with Diabetes: The Central Role of the Cardiomyocyte

**DOI:** 10.3390/ijms20133299

**Published:** 2019-07-05

**Authors:** Tiziana Filardi, Barbara Ghinassi, Angela Di Baldassarre, Gaetano Tanzilli, Susanna Morano, Andrea Lenzi, Stefania Basili, Clara Crescioli

**Affiliations:** 1Department of Experimental Medicine, “Sapienza” University, Viale del Policlinico 155, 00161 Rome, Italy; 2Department of Medicine and Aging Sciences, “G. D’Annunzio” University of Chieti and Pescara, Via dei Vestini 31, 66100 Chieti, Italy; 3Department of Cardiovascular Sciences, “Sapienza” University, Viale del Policlinico 155, 00161 Rome, Italy; 4Department of Translational and Precision Medicine, “Sapienza” University of Rome, Viale del Policlinico 155, 00161 Rome, Italy; 5Department of Movement, Human and Health Sciences, Section of Health Sciences, University of Rome “Foro Italico”, Piazza L. de Bosis 6, 00135 Rome, Italy

**Keywords:** diabetes, cardiomyopathy, cardiomyocytes, chemokines, inflammation, therapy

## Abstract

The term diabetic cardiomyopathy (DCM) labels an abnormal cardiac structure and performance due to intrinsic heart muscle malfunction, independently of other vascular co-morbidity. DCM, accounting for 50%–80% of deaths in diabetic patients, represents a worldwide problem for human health and related economics. Optimal glycemic control is not sufficient to prevent DCM, which derives from heart remodeling and geometrical changes, with both consequences of critical events initially occurring at the cardiomyocyte level. Cardiac cells, under hyperglycemia, very early undergo metabolic abnormalities and contribute to T helper (Th)-driven inflammatory perturbation, behaving as immunoactive units capable of releasing critical biomediators, such as cytokines and chemokines. This paper aims to focus onto the role of cardiomyocytes, no longer considered as “passive” targets but as “active” units participating in the inflammatory dialogue between local and systemic counterparts underlying DCM development and maintenance. Some of the main biomolecular/metabolic/inflammatory processes triggered within cardiac cells by high glucose are overviewed; particular attention is addressed to early inflammatory cytokines and chemokines, representing potential therapeutic targets for a prompt early intervention when no signs or symptoms of DCM are manifesting yet. DCM clinical management still represents a challenge and further translational investigations, including studies at female/male cell level, are warranted.

## 1. Introduction

Diabetic cardiomyopathy (DCM) ending in left ventricular (LV) dysfunction is reported to be the leading cause of death in type 2 diabetes (T2D) [1]. Patients suffering from T2D retain a 2- to 5-fold higher risk of developing cardiovascular disease vs. a non-diabetic matched population, as a consequence of diabetes-associated coronary atherosclerosis and vascular abnormalities [2]. However, regardless of the common co-morbidities present in T2D patients, it has been observed that DCM occurs independently of vascular diseases, as coronary artery disease or hypertension [3,4,5]. The term DCM was first coined in 1972 by Rubler and collaborators, who observed congestive heart failure without any alteration of coronary arteries or valves in a very restricted number of patients [6]. Nowadays, albeit some debate still exists whether or not DCM alone can frame a clinical discernible picture [2], it is accepted that diabetes per se is able to trigger a wide range of biomolecular changes that remodel cardiac tissue and cells, with a unique pattern characterizing DCM vs. other types of cardiomyopathies [7,8]. Hyperglycemia, dyslipidemia and hyperinsulinemia related to the diabetic insult represent critical events influencing cardiac (worse) outcome. Although the aetiology is different, DCM can occur also in type 1 diabetes (T1D); high glucose-dependent metabolic alterations, oxidative stress and inflammation constitute the common features at the initial phases of the disease, suggesting a possible unifying hypothesis [9].

Indeed, the toxicity associated with disturbances in circulating levels of glucose and insulin (I) or fatty acids (FA) leads to alterations of cardiac structure, cardiomyocyte signaling and metabolism [10]—tissue fibrosis, myocyte cell death, contractile dysfunction and oxidative stress, to mention some. Since hyperglycemia is the major factor in DCM etiology, an optimal glycemic regulation is undeniably the first step to limit glucose-induced toxicity. However, beyond glycemic status, the evidence that some drugs currently used to control glycemia exert some important beneficial effects directly onto myocardium suggests the importance of local biomolecular factors at tissue/cell level [11,12]. Furthermore, the glycemic status, albeit eliciting critical effects onto cardiac cell function, seems not to be predictive of early heart function decline [13]. The hypothesis that the failure of heart function in T2D could occur following early alterations within the cardiac cell has been put forward [14].

The challenge for clinicians and researchers is to clarify the early events occurring at cardiomyocyte level preceding heart remodeling and geometrical changes in tissue architecture, which represent the final consequence of a series of detrimental events related to chronic hyperglycemia [15]. Cardiac cell death is hypothesized to be the basic event initiating cardiac remodeling within a T helper 1 (Th1)-type inflammatory microenvironment. In this scenario, some biomolecules with high immune-activity, like Th1-type cytokines and chemokines, play pivotal roles in the mechanisms underlying DCM onset and progression. Thus, despite the undeniable role of vascular bed-related changes, cardiac cells are not considered only final “passive” targets of detrimental events.

This paper aims to focus on the cardiomyocyte, which, under chronic hyperglycemia, behaves as an active unit, critically participating in and contributing to the immune/inflammatory dialogue between local and systemic counterparts underlying DCM onset, development and maintenance. Some of the main biomolecular mediators and mechanisms at cellular and intracellular level in cardiac cells will be overviewed and discussed also as potential new pharmacologic targets.

## 2. DCM Etiology: The Pivotal Role of the Cardiomyocyte at Disease Onset

The major etiological factor in DCM development is hyperglycemia, which is responsible for disease stepwise progression [16]. Four possible stages have been reported: from the asymptomatic initial phase, clinically marked by LV hypertrophy with preserved ejection fraction (EF); followed by reduced EF and dilatation at stage 2; by systolic and diastolic dysfunction, micro-angiopathy, hypertension and myocarditis at stage 3; culminating in end-stage or refractory heart failure with ischemia, infraction and remodeling in stage 4 [5]. The overall chronic exposure to hyperglycemic milieu is tightly associated with extended organ and tissue injury, including micro- and macrovascular damages, nephropathy and neuropathy, due to either vascular-mediated action or direct effects of hyperglycemia on tissues. However, a full description of clinical stages and hyperglycemia-dependent damage on organs would be very extensive and it is beyond the aim of this review. The following paragraphs summarize how early metabolic abnormalities and inflammatory perturbation within cardiomyocytes eventually leads to heart dysfunction in DCM. 

## 3. Cardiomyocyte Metabolism in Normal and Diabetic Condition

### 3.1. Metabolic Substrate Flexibility

The cardiomyocyte with its continuous contraction retains the highest demand for energy and shows a unique substrate promiscuity, which allows the cell to utilize multiple substrates, such as FA, carbohydrates, aminoacids, lactates and ketons [17,18], for ATP and energy production. The ability to utilize such substrate variety under normal conditions is reported as “metabolic substrate flexibility” [12], which is lost when heart is under hyperglycemic/hyperinsulinemic milieu. Normally, FA and glucose are the main utilized substrates by cardiac cells. FA exogenous supply, deriving from the lipolysis of circulating complex lipids or released by adipose tissue, occurs through a number of transporters (CD36 or FA translocase, FA binding protein or FABP_PM_, FA transporter proteins or FATP); an endogenous FA source is from cardiac triglycerides storage. FA enter oxidative cycle by conversion in acyl coenzyme A (acyl-CoA), which, once within the mitochondria, undergoes β-oxidation, leading to ATP. Glucose uptake is mainly an I-dependent event mediated first by GLUT4 translocation through I receptor substrate (IRS)1/2, and downstream by PI3K/protein kinase B (Akt) [18] and AMP-activated protein kinase (AMPK). The process ends in glycogen storage or ATP production through glycolitic and oxidative processes, via pyruvate production in the cytoplasm (controlled by phosphofructokinase) and ATP formation from pyruvate oxidation (controlled by pyruvate dehydrogenase) within mitochondria. Part of FA or glucose can also enter into biosynthetic paths [19,20]. During normal aerobic perfusion and workloads, FA oxidation accounts for about 70% of energy generation; periods of ischemia or overload works switch substrate utilization to glycolysis and pyruvate oxidation [21].

It has been known for a while that hyperglycemia and insulin resistance (IR) significantly reduce GLUT4 recruitment and deplete glucose uptake [22]. Accordingly, a significant reduction of GLUT4 expression and PI3K/Akt signaling has been described in LV cardiac biopsies from T2D patients [23]. Simultaneously with reduction of glucose utilization, an increase in FA supply, β-oxidation and internalization/deposition of FA intermediates within cardiomyocytes, occurs [24]. Cardiomyocytes are not properly equipped for lipid storage; hence, FA oxidation-dependent lipotoxicity occurs and further impairs glucose oxidation (limiting glucose and pyruvate utilization by pyruvate dehydrogenase inhibition), triggers cell apoptosis (through peroxisome proliferator-activated receptor/PPAR) [21] and inhibits autophagy [25,26,27]. In this condition, heart metabolic flexibility is lost together with an efficient ratio between ATP production and substrate utilization [25]. The overall unbalance between increased oxygen mitochondrial consumption and reduced cardiac bioenergetic efficiency [28,29] ends in perturbation of contraction/relaxation coupling [30]; meanwhile, defective excitation/contraction coupling goes together with molecular changes in sarcoplasmic reticulum, which is responsible for the impairment in cardiomyocyte calcium handling, as observed in cardiac diastolic dysfunction at DCM onset [15]. So far, the hyperglycemia-induced shift towards an increased metabolic dependence on FA oxidation in mitochondria is seen as the primary injury within cardiomyocytes in DCM pathogenesis [30], which is independent of the hyperlipidemia effect on coronary endothelium [21].

### 3.2. Advanced Glycated End Products (AGE), Renin-Angiotensin-Aldosterone System (RAAS), Damage-Associated Molecular Pattern (DAMP) and Cardiomyocyte Damage

To provide alternative energy sources, which coincide with GLUT4 and glucose uptake downregulation, the expression of I-insensitive transporters for galactose and fructose has been shown to increase [31,32] and trigger fructose intracellular accumulation, likely through sorbitol path overactivation [33,34,35]. In turn, sorbitol accumulation is accompanied by reactive oxygen species (ROS) generation/ROS scavenger reduction on one side, and by DNA fragmentation and cell shrinkage dependent on hyperosmolarity on the other side. Other sources of ROS production have been described within the heart [4,36]; the overall excess in ROS amount is associated with cardiac oxidative stress development, ending, i.e., in protein, lipid and DNA extended damage, autophagy dysregulation and nitric oxide (NO) reduced bioavailability (necessary for an optimal function [18,37]), which is a hallmark of DCM [6,38]. Impaired NO production, besides important effects onto coronary vasculature—i.e., impaired vessel relaxation and capillary recruitment—promotes cardiac stiffness by enhancing collagen crosslinking enzymes [6,39,40]. Moreover, high glucose level-induced formation of AGE, deriving from non-enzymatic binding between sugars and protein/lipid amine residues, cross-link and slow collagen molecule turnover [41], further contribute to fibrotic processes. In fact, fibrosis of a diabetic heart is accompanied by extracellular matrix (ECM) abnormalities in protein structure and turnover, and in collagen deposition. AGE/RAGE (receptor for AGE) interaction also promotes oxidative stress and pro-fibrotic mediators through some intracellular path activation, such as Janus kinase (JAK), mitogen -activated protein kinase (MAPK) or transforming growth factor β1/SMAD [15]. 

Another factor contributing to hyperglycemia-induced heart damage is the abnormal activation of local RAAS that induces functional abnormalities within ventricular myocytes, culminating in fibrosis and necrosis [42,43,44,45]. The release and activation of proinflammatory mediators, particularly ROS or AGE, and the DAMP represents potent triggers of inflammatory processes, critically contributing to the so-called cardiac maladaptive proinflammatory response. In this scenario, we point out how not only does the cardiomyocyte behave as cellular target of biomolecules released by immune cells, but it also acts as an immune-active unit participating in the detrimental process of DCM onset and maintenance.

## 4. Cardiomyocyte Inflammation 

There are growing lines of evidence indicating myocardial inflammation as a key process in DCM development [7,46,47,48]. While myocardial inflammation per se represents an “adaptive” early response to restore homeostasis against short-term stress and abnormal conditions [46,47,48,49], it turns into a “maladaptive” proinflammatory response under persistent stressful challenge, like diabetes. Indeed, chronic low-grade inflammation, also named para-inflammation/meta-inflammation [47], is a subclinical condition recognized to be the pathogenic event within cardiomyocytes exposed to lipid and sugar excess, as depicted in Figure 1.

### 4.1. The Inflammasome Platform

Glicotoxicity and lipotoxicity exert deleterious effects at subcellular level by upregulating the expression of a multiprotein signaling complex regulator of inflammatory processes and cellular death, which is the nucleotide-binding oligomerization domain like receptor pyrin domain containing 3 (NLRP3) inflammasome. In T2D, consequent to the aforementioned metabolic substrate flexibility shift towards lipid utilization, there is ROS overproduction associated with DAMP [50], which, among other events, promotes NLRP3 expression (first priming) and structural modulation for inflammasome platform assembly (second step activation). In particular, high glucose-induced ROS engage nuclear factor-κB (NF-κB) and thioredoxin interacting/inhibiting protein (TXNIP), respectively, mediate the first signal for inflammasome induction and the second signal for inflammasome complex oligomerization [50,51,52,53,54]. Albeit clear evidence of direct activation of intracellular NLRP3 by lipotoxicity is still lacking, intracellular lipid accumulation has been documented within T2D cardiomyocytes. In turn, inflammasome activation is associated with interleukin (IL)-1β and IL-18 production, which both induce cardiomyocyte apoptosis, the first step initiating DCM structural remodeling [55,56]. Indeed, IL-1β and IL-18 are substrates of caspase-1 that, following NLRP3 platform assembling, cleaves the cytokines to their mature/bioactive form and triggers pyroptosis, a process causing cell membrane pore formation/rupture and cell death [57,58]. In line with this observation, tissue biopsies from DCM hearts show 85-fold more apoptosis than non-diabetic hearts [59]. Although the underlying mechanisms are not yet fully clarified, discovering NLRP3 biomolecular function gives a mechanistic explanation for cytokine-mediated initiation and the progression of DCM [60,61].

### 4.2. Leukocyte Infiltration in the Damaged Cardiomyocyte

So far, understanding the early mediators of low-grade inflammation at cellular and subcellular levels within the heart, when morphological or clinical signs of DCM have not yet manifested, captured the attention of researchers. As addressed, in DCM several hyperglycemia-induced processes cause tissue/cell damage, thus driving the infiltration and accumulation of leukocytes, i.e., T lymphocytes, neutrophils and activated macrophages, to the lesion sites. Upon infiltration, immune cells contribute in promoting local inflammation through the activation of specific biomolecular processes and mediators, as briefly summarized herein. Neutrophils, which normally represent the first-line of defense involved in tissue repair by polarizing macrophages towards a reparative phenotype (M2) [62], seem to contribute to DMC development through DNA and granule protein release that prime other immune cells and amplify inflammation. Those processes involve neutrophil extracellular traps (NETs) formation and release, which, indeed, is enhanced in diabetic patients [63,64]. Albeit the fine-tuning mechanisms are still to be described, NET release or NETosis participates in cell death regulation via the formation/activation of some biomolecules and paths, such ROS and NF-kB, and, when dysregulated, provides strong proinflammatory stimuli [64,65]. 

Activated macrophages usually reduce inflammation during tissue injury by efferocytosis, a phenomenon based on apoptotic cell whelm and cellular debris phagocytosis [66]. Phagocytic, lysosomial and chemotactic activity [67,68] of macrophages are impaired by hyperglycemia in diabetic patients and correlate with blood glucose level [69]. The classical subdivision of macrophages in phenotype M1 promoting chronic tissue inflammation and IR [70] and pro-reparative M2 phenotype seems, nowadays, overcome by the hypothesis of multiple macrophage phenotypes [71]—still to be elucidated. However, even if macrophage impact on DCM development remains largely unknown, M1 type cells are predominate in diabetes and are upregulated in the myocardium before cardiac dysfunction [72].

The role of T cells is clearly documented in cardiac injury; i.e., tissue fibrosis and LV function are protected by T cell depletion [73,74,75,76]. Th and T regulatory (Treg) subsets significantly participate in inflammation, IR and vascular alteration in T2D. In particular, the contribution of Th1, Th17 and Th22 pro-inflammatory subtypes to coronary artery disease onset has been reported in DCM patients—after adjusting for age, sex and disease duration—along with a simultaneous decrease of the pro-regulatory anti-inflammatory Th2 subtype [77]; Treg cells with a suppressor function on Th1 are decreased as well. Consequently, the alteration of the Treg/Th17 and Treg/Th1 ratios in favor of pro-inflammatory subsets occurs in T2D patients [78]. Treg cells, indeed, can control inflammation and tissue impairment by suppression of Th proinflammatory phenotypes, i.e., Th1 and Th17, through several biomolecular mechanisms, especially involving cytokine release or suppression [79,80]. Although further explorations are mandatory to elucidate role(s) and mechanism(s) of action of lymphocyte subsets, it is undeniable that they critically participate in DCM context through cytokines.

## 5. The Cytokine Hypothesis

The hypothesis that proinflammatory cytokines play a pivotal role in the general heart failure setting, in addition to hemodynamic disorders and neurohormones, has been proposed by Seta et al. since 1996 [81]. The so-called “cytokine hypothesis” assumes that cardiac disease progression and heart failure, regardless of the etiology, is due to a deleterious cytokine cascade established in heart cells and peripheral circulation. Cytokines, indeed, represent a portfolio of immunoactive molecules mediating inflammatory processes converged on cardiac cells to reestablish the balance when homeostasis is altered, regardless of the cause. When this kind of “switch-on response” fails to restore cardiac homeostasis and persists—i.e., under chronic hyperglycemia—it would end in sustained inflammation and cytokine overproduction, the latter mirroring the maladaptive response of cardiac cells. Whether cytokines really represent the inability of cardiomyocytes to restore homeostasis remains speculative; however, the presence of proinflammatory cytokines in a failing heart has been documented for a while [82]. Indeed, a plethora of these molecules are expressed in LV dysfunction when still asymptomatic and intervene in heart wall stiffening and decreased contractility [83,84,85], i.e., TNFα, TGF-β, IL-6, IL-1β and interferon (IFN)γ [86,87,88], which are released by macrophages and lymphocytes at inflammation onset, seeming to initiate or worsen cardiac injury. IL-1β and IL-18 promote fibroblast phenotype and apoptosis [55,56], the TNF superfamily mediates collagen degradation and progressive LV dilation [89], IL-6 is involved in tissue injury and heart failure [84] and soluble ST2 (sST2), the receptor for IL-33, is the first inflammatory prognostic biomarker approved by the Food and Drug Administration (FDA) for heart failure [47]. 

Pro-inflammatory cytokines are expressed by nucleated cell types residing/infiltrating the myocardium [82]. Those cytokines released locally, in turn, would exacerbate tissue injury by promoting vascular permeability and further leukocyte invasion through autocrine or paracrine mechanisms [90,91,92]. Higher blood levels of inflammatory cytokines, i.e., TNFα or IL-6, correlate with cardiac disease severity and worse prognosis [82,93,94], with stronger impact on subjects affected by diabetes or metabolic syndrome [95]. 

To date, among the different cell types within the heart, which contain the endothelial cells, vascular smooth muscle cells, fibroblasts, myocytes and immune cells, all contribute to functional maintenance or disease development, and emerging attention has been addressed towards the complex dialogue between cardiomyocytes and inflammatory cells.

While there is evidence that the failing heart releases pro-inflammatory “cardiokines,” the specific contribution of cardiomyocytes still remains to be elucidated [96,97]. Nevertheless, among the several biomediators released by cardiac cells upon inflammatory challenge, chemokines emerged to play critical roles. 

## 6. The Chemokines

The chemokines, or *chemo*attractive cyto*kines*, are small molecules (about 70 aminoacidic residues) originating from one ancestral gene (about 650 million years ago) [98] with powerful chemoattractant activity.

Briefly, chemokines are classified in four main subtypes, CC, CXC, C and CX3C chemokines, depending on the space separating their first two cysteine residues (indicated as “C”) [99]. This superfamily represents an important class of biomediators in the cross-talk among immune, cardiovascular and autonomic nervous systems in health and disease, attracting attention as biomarkers with the ablity to predict cardiovascular events even in healthy subjects. The whole chemokine family is constantly growing, as new components are discovered; now it consists of fifty-four members, nineteen of which are shown to be relevant in different cardiac diseases. Whereas a full description of chemokine structure/classification/function is beyond the aim of this paper and extensively treated elsewhere [100], herein it is relevant to highlight their ability to act in a “working-network” instead of “one chemokine-one function” mode, in response to cardiac injury, regardless of the cause. This modality, indeed, likely reflects the typical redundancy observed in inflammatory processes and it is quite clear that it potentially represents a significant opportunity for novel therapeutic approaches to control inflammation underlying cardiac diseases.

### CXCL10 and CXCL8 Potential Therapeutic Targets of PDE5i in DCM

We have reported on two chemokines as potential therapeutic targets in DCM, the interferon γ-induced 10 kD protein IP-10, or CXCL10, and IL-8, or CXCL8 [101,102]. They both belong to the CXC family, sub-grouped as chemokines ERL−, as in CXCL10, or ERL+, as in CXCL8, according to the absence/presence of Glut-Leu-Arg motif, which confers angiostatic or angiogenic properties, respectively [103]. An additional classification in two main functional categories is reported: the inflammatory/inducible chemokines, participating to effector leukocyte recruitment at injured sites, and the homeostatic/constitutive chemokines, dedicated to immune surveillance, lymphocyte and DC trafficking. A further third group referred to as “dual function” chemokines, impossible to be unambiguously assigned to the previous categories, is engaged in immune defense functions and targets also non-effector leukocytes, such as CXCL10 [104,105]. 

Briefly, CXCL10 regulates and controls different biological responses in physiologic and pathologic conditions [99]. It is secreted by leukocytes and some type of tissue cells under inflammatory stimuli, including human endothelial cells, vascular smooth muscle cells, fibroblasts, keratinocytes, skeletal muscle cells and cardiomyocytes [106,107,108,109]. Upon binding with its receptor CXCR3, mainly expressed by T cells, natural killer (NK) and monocytes—and, at low level, by endothelial and vascular smooth muscle cells [110]—acts as a potent chemoattractant, responsible for leukocyte trafficking from the bloodstream to the site of tissue injury or inflammation. Of note, this chemokine is not linked to generic inflammatory status and is the first one triggering early inflammatory response in several processes [99]. Since the pioneering studies in heart transplantation [111], CXCL10 has been identified as the only chemokine induced by isogenic tissue transplantation and the first ligand detected after allograft transplantation [112]. Its neutralization resulted in a prolonged graft surviving and decreased rejection in cardiac and multiorgan transplant models [99], suggesting the pivotal role of the CXCL10-CXCR3 axis in early Th1-driven response. Then, multifaceted functions of CXCL10 in several cardiovascular diseases [110], such as atherosclerosis and plaque formation, aneurysm, infarction, myocarditis and cardiopulmonary bypass have emerged [113,114,115,116].

CXCL8 also participates in the early inflammatory stages targeting neutrophils, which, through granule enzyme release, are responsible for tissue re-arrangements and degradation [117]. Differently from many proinflammatory mediators cleared within hours after synthesis and release, CXCL8 continues to be active for long time [118] and contributes to vascular dysfunctions, like atherosclerosis, aortic aneurysm formation and hypertension [119]. Interestingly, it takes part in cardiac diseases (primary or dysmetabolism-associated [120]) and correlates with baseline cardiovascular risk [121]. CXCL8 high blood level in T2D patients seems associated with worse inflammatory and cardiometabolic profile [122]. Higher baseline levels of CXCL10 and CXCL8 as detected in patients with different cardiac diseases vs. healthy subjects likely mirror early inflammation underlying cardiac damage, potentially reflecting different levels of disease stage and severity.

Usually standard biomarkers used to determine myocardial injury, i.e., pro-brain natriuretic peptide or troponin I, are detectable after tissue injury, thus reflecting already established damage(s). Thus far, many clinical studies attempt to define an early assessment of cardiac risk related to chemokine circulating levels. Although there are some controversial data, CXCL10 seems a good independent and stable predictor of coronary heart disease (CHD) [123], heart rejection [99,124,125,126] or other cardiovascular events or death [127,128,129,130].

Furthermore, some studies in animal models also report that fractalkine or CX3CL1, a (C-X3-C motif) chemokine, is expressed at early stages of diabetes in both cardiomyocytes and cardiac fibroblasts (in addition to MCP-1).

Notably, soluble fractalkine, besides acting as a potent chemo-attractant, can directly affect cardiomyocytes contractile machinery by binding its receptor on heart cells, leading to a decreased speed of contraction and relaxation both under basal conditions and beta-adrenergic stimulation [131,132]. 

However, instead of single chemokine approach, multiple chemokine assessment is strongly recommended in relevant biomarker analysis finalized for prognostic/diagnostic purposes. In line with literature, we also analyzed and observed high blood levels of CXCL10 and CXCL8 in DCM patients and we investigated them as potential pharmacological targets at systemic and local level, rather than disease predictors. As from our research in 46 diabetic subjects, enrolled in a trial [102] (Clinical Trial Registration—URL: http://www.clinicaltrials.gov. Unique identifier: NCT00692237) at the initial stage of DCM, blood level of CXCL10 and CXCL8 was significantly reduced after three month treatment with the phosphodiesterase inhibitor 5 (PDE5i) sildenafil (100 mg/day), vs. placebo; at the same time, whereas drug intake modified metabolic parameters (hemoglobin A1c, post-prandial glycemia and lipidemic profile [133]), it failed to change cardiac standard markers (ejection fraction, mass and volume index or blood pressure). This effect was not so surprising considering that all patients were at DCM onset and showed preserved LV function and no signs of ischemia. In this light, we hypothesize that PDE5i likely acts on and declines early inflammatory biomediators like CXCL10 and CXCL8 before clinical signs manifest, offering a precious opportunity to intervene on disease development in the initial temporal frame. Remarkably, in human isolated cardiomyocytes, both chemokines, virtually absent in basal conditions, significantly increased after Th1 stimuli. While CXCL10 secretion significantly declined after sildenafil, the PDE5i failed to decrease CXCL8, which decreased with mycophenolate and cyclosporine A [101].

Thus far, human resident cardiomyocytes under inflammatory environment behave as immunoactive cells and likely contribute to chemokine blood rise, perpetuating the vicious detrimental loop through enhancement of immune cell infiltration. Some of the detrimental processes are summarized in Figure 2.

And, importantly, as immuneactive units, human Th1-inflammation activated cardiomyocytes are able to respond to PDE5i and to some immunomodulating drugs [101,134,135]. 

## 7. An Overview of the Anti-Inflammatory Approach in DCM Treatment

It is well accepted that PDE5i as sildenafil are vasoactive drugs commonly used to treat erectile dysfunction (ED), showing interesting “extra-erectile” effects based on their intrinsic anti-Th1 inflammatory action, exerted through cGMP/cAMP stabilization [136,137,138]. Accordingly, PDE5i-induced protective effects are described in several cardiac diseases, from heart failure to ischemia/reperfusion injuries, infarct, ventricular arrhythmias and cardiopulmonary bypass [139,140,141,142,143]. Of note, PDE5i use strongly associates with reduced mortality rate and hospitalization in a cohort of T2D patients at high cardiovascular risk [144], likely enhancing antioxidant enzyme system [145]. In light of this evidence, it has been hypothesized that PDE5i-elicited cardioprotective effects depends on their anti-Th1 inflammatory activity, besides their vasoactive action. In line with this hypothesis, with progress in understanding the role of inflammation during DCM development, early Th1-driven processes/paths/mediators have been considered therapeutic targets for early interventions aimed to prevent or delay DCM onset and progression.

As previously addressed, the effectiveness of glucose lowering agents in reducing cardiac events in T2D is still to be demonstrated, as from the results of two randomized intervention trials (Action in Diabetes and Vascular Disease: Preterax and Diamicron MR Controlled Evaluation (ADVANCE) and Action to Control Cardiovascular Risk in Diabetes Study (ACCORD)), which reported an increased mortality [146,147] when considering the number of variables to be included—i.e., level of co-morbidity, age, disease duration and baseline glycated hemoglobin. Thus, the cardiovascular outcome became a requirement in trials on anti-hyperglycemic molecules. Among anti-diabetes drugs, empaglifozin, belonging to the kidney-targeted sodium glucose cotransporter 2 inhibitors (SGLT2i) class, exhibited a significant improvement in cardiovascular outcome, reducing cardiovascular events (14%), heart failure-related hospitalization (35%), cardiovascular mortality (38%) and all-cause death (32%) in T2D patients at high cardiovascular risk ((Empagliflozin) Cardiovascular Outcome Event Trial in Type 2 Diabetes Mellitus Patients (EMPA-REG OUTCOME)) [148,149]. Similar encouraging results were obtained in other trials on the SGLT2i canaglifozin (CANagliflozin cardioVascular Assessment Study (CANVAS)) [150]. SGLT2i-induced beneficial effects seem to rely on reduction of inflammation, oxidative stress, endothelial damage and heart remodeling, independently of glucose control [151]. Accordingly, glycated hemoglobin, body weight, circulating I and blood pressure modestly decreased after SGLT2i in comparison to the standard glucose-lowering therapies metformin, I and sulfonylureas. Further investigations on canagliflozin, empagliflozin and dapagliflozin (Comparative Effectiveness of Cardiovascular Outcomes in New Users of SGLT-2 Inhibitors (CVD-REAL)) strengthened the hypothesis that cardiac benefits are linked to a class effect rather than to a specific molecule [152]. Despite the lack of myocardial SGLT-2 expression in the human heart and the lack of clarity in SGLT-2i mechanism(s) of action [12], this class of drugs improves myocardial metabolism [153], increases ATP level, enhances cardiomyocyte viability and suppresses inflammation, i.e., targeting NLRP3 inflammasome, lowering Na^+^ and activating AMPK, as extensively described elsewhere [151,152]. Activation of AMPK, in particular, seems to prevent DCM progression, representing, therefore, a promising therapeutic target [15].

Given the importance of inflammation, cytokine antagonism/suppression have been proposed as a novel approach for DCM treatment since it reduces intramyocardial inflammation and cardiac fibrosis in animal models [154]. Trials on TNF or IL-1 neutralization with monoclonal antibodies or receptor antagonists, respectively, resulted in controversial data or even in worse outcomes—i.e., etanercept, a TNF soluble receptor to block TNFα, which, in some cases, acts as an agonist [155,156]. Therapeutic intervention based on anti-inflammatory, immunesuppressants (methotrexate) or immunomodulators (Immune Modulation Therapy, IMT), the latter especially downregulating IL-8 and IL-1β, gave disappointing results or some important side-effects, like a compromised host defense or secondary compensatory inflammatory processes [47].

In this scenario, while the question arose on the use of “straight” immunomodulating drugs to control Th1 processes in DCM, the recommendation of relatively “safe” drugs like PDE5i with a very good tolerance profile, exerting beneficial effects onto metabolic and inflammatory status, might be foreseeable. In addition, sildenafil left intact cardiac cell viability, at variance with other immunosuppressants like mycophenolic acid [134].

## 8. A Window Opening on Sex-Dependent Molecular Mechanisms in the Cardiomyocyte Developing DCM

It is a fact that DCM and cardiovascular risk are higher in females, even though T2D incidence is about the same for men (6.6%) and women (5.9%) [157]. Whereas higher estrogen levels play a protective role on the cardiovascular system of females, as estrogens drop off in menopause and post-menopause or in presence of T2D, the female advantage is lost [158]. Indeed, premenopausal non-diabetic women show cardiovascular events about ten years later than men, but remarkably, the female advantage is lost in presence of T2D [158]. From the Framingham Heart Study, a number of epidemiological and associative studies highlighted the role of estrogens indicating DCM as a sex- and age-dependent event [159], however the research on estrogen-dependent intracellular signaling within the female myocardium is still in its infancy. Nevertheless, from in vitro and animal investigations, there is mounting evidence on some remarkable sex differences at the biomolecular level between female and male cardiomyocytes progressing from health to DCM condition. As examples, data from experimental rat models show in female hearts higher expression of some microRNA (miRNA) associated with T2D and heart dysfunction (i.e., miR-208a) and downregulated by estrogens [160,161,162]. Differently from females, during DCM development, male cardiomyocytes show an increase in collagen, fibrosis, fatty acid uptake and some cytokine decrease (IL-2, IL-10 and IFNγ) [163].

Thus far, promising data on sex-dependent mechanisms emerge at cardiomyocyte level as well, albeit we cannot ignore that preclinical experimental data in animal models and cells present important limitations when compared to studies in humans and cannot be straightforwardly translated due to specie-specific bias [163]. To date, topics on miRNA and heart failure and preserved ejection fraction in women have been recently reviewed [164].

## 9. Conclusions

Nowadays, it is accepted that DCM, the leading cause of mortality among diabetes-associated macrovascular complications, develops from Th1 type driven biomolecular and functional modifications within the cardiomyocyte. Maladaptive proinflammatory processes represent the response of cells challenged by high glucose; unfortunately, even an optimal glycemic control seems not enough to counteract DCM. Instead, recognizing the early Th1 type processes and biomediators, such as inflammasome platform or chemokines within cardiomyocytes following glycemic excursion, might result in novel strategies for early interventions aimed to prevent or delay DCM progression. 

Furthermore, in line with data from clinical studies, there is emerging evidence documenting sex-dependent differences between female and male cardiomyocytes, in health and DCM.

Albeit the progress achieved in basic research, DCM remains largely unknown for several limitations, i.e., the use of animal models often cannot warrant the leap into human trials; dealing with human cardiomyocytes for in vitro experiments is not always possible since these are post-mitotic cells with a limited lifespan or biological limits when cultured or immortalized with viral vectors. Sex-related differences observed not only in women/men outcomes but, remarkably, at female/male cell level, might not be limited to sex hormone milieu. Further studies with multifaceted translational approaches are mandatory to identify the timeframe for early fine-tuned interventions in men and women aimed to prevent/manage DCM.

## Figures and Tables

**Figure 1 ijms-20-03299-f001:**
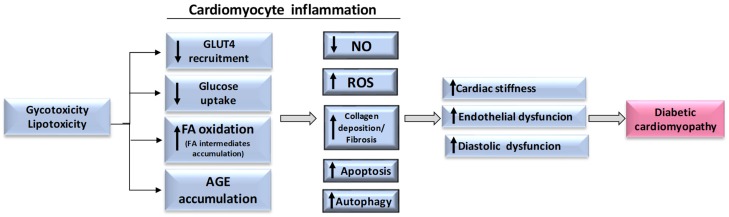
Glycotoxicity/lipotoxicity-induced events at cardiomyocyte level. A “maladaptive” proinflammatory response occurs in cardiomyocytes under persistent stressful challenge, like diabetes. Each box depicts intracellular processes leading to diabetic cardiomyopathy. FA: fatty acids; AGE: advanced glycated end products; NO: nitric oxide; ROS: reactive oxygen species.

**Figure 2 ijms-20-03299-f002:**
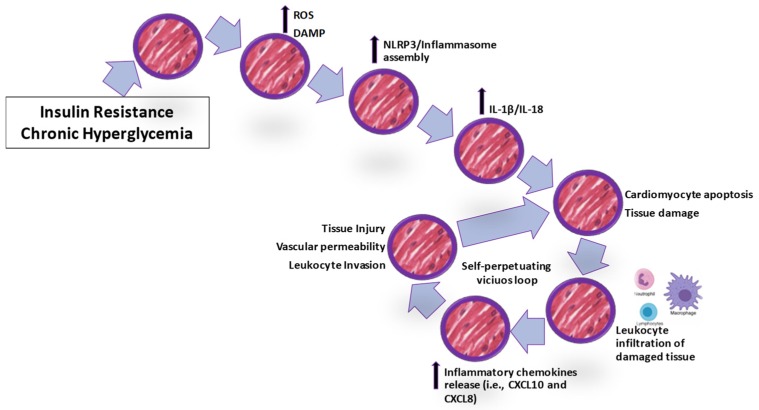
Detrimental event cascade leading to cardiac tissue injury. Chronic hyperglycemia triggers oxidative stress mediators and inflammasome assembly, leading to cell apoptosis and the release of cytokines and chemokines, which perpetuate inflammation. ROS: reactive oxygen species; DAMP: damage-associated molecular pattern; NLRP3: nucleotide-binding oligomerization domain like receptor pyrin domain containing 3. IL: interleukin.

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
