# Peer review of "Cardiomyopathy Associated with Diabetes: The Central Role of the Cardiomyocyte"

_ijms, 2019, doi:10.3390/ijms20133299_

Round 1

Reviewer 1 Report

The review represents an update of the role of inflammation in the development of diabetic cardiomyopathy, highlighting the involvement of specific classes of mediators of inflammation inducing cardiac damage.

The perspective of heart dysfunction under diabetes as an immunometabolic disorder is very interesting and of potential interest for diabetes-induced cardiomyopathy management.

I only suggest the correction of some writing mistake and the punctuation of the manuscript:

Page 4 –(necessary for an optimal function [17,36], which is a hallmark of DCM”: Please, add the close bracket.

Page 8 – “(hemoglobin A1c, post-prandial glycemia and lipidemic profile [127]”: Please, add the close bracket.

- “It failed to change cardiac standard makers”: are they “makers” or “markers”?

Page 9: “to respond respond”: please, delete “respond”.

Author Response

We thank the Reviewer 1 for the comments. All indicated writing mistakes have been corrected.

In particular:

-       the closed bracket has been added after the references in the sentence: “(necessary for an optimal function…)”, page 4, bottom line;

-       the closed bracket has been added to after the reference in the sentence: “(hemoglobin A1c, post-prandial glycemia and lipidemic profile…)”, page 8, sixth line from the bottom;

-       “makers” has been corrected in “markers”, page 8, fifth line from the bottom,

-       “respond” has been deleted, page 9, last line of subparagraph 6.1.

Misspelling and mistakes have been corrected throughout the text.

Reviewer 2 Report

The Authors provide an interesting review showing the central role of cardiomyocytes in the development of diabetic cardiomyopathy. All the observations reported in the manuscript further underline that hyperglycemia can affect directly myocardial tissue giving rise to metabolic changes and pro-inflammatory cytokine/chemokine expression.

As the Authors stated, the identification of citokines and chemokines produced by cardiac cells, capable of altering cardiomyocyte function and myocardial diabetic microenvironment, could represent a therapeutic target to prevent the occurrence of the overt DCM phenotype.

The paper is clear and well written and offers a comprehensive overview of the topic considered. I have only the following  suggestions.

Although the aetiology is different,  diabetic cardiomyopathy can occur in both type 1 and type 2 diabetes. Hyperglycemia, metabolic alterations, oxidative stress  and inflammation constitute common features at the initial phases of the disease, suggesting  a possible unifying hypothesis (Seee also Poornima IG, Circ Res 2006). Can the Authors include a sentence in the introduction section on this point?

Chapter 6 and subheading 6.1 relating to chemokines: In addition to the chemokines cited,  I suggest to take into consideration the subtype CX3C and specifically CX3CL (Fractalkine) already known to be expressed,  at early stages of diabetes, from both cardiomyocytes and cardiac fibroblasts (in addition to MCP-1), in experimental models of diabetes.  This is an important finding, by considering that soluble Fractalkine, besides acting as a potent chemo-attractant can directly affect cardiomyocyte contractile machinery, by binding to its receptor on cardiomyocytes, leading to a decreased speed of contraction and relaxation under basal conditions, as well as under beta-adrenergic stimulation (Taube, D et al. PLoS ONE 2013; Savi M et al., Nutrients 2016).

Author Response

We thank the Reviewer 2 for the suggestions which improve the manuscript.

In particular:

-       A comment on DCM occurring also in type 1 diabetes and on possible unifying hypothesis with inflammation as a common feature between T1D and T2D has been added in Introduction; the related reference quoted as new number [9] has been added;

-       A topic on CX3CL1/fractalkine has been addressed in chapter 6 and subheading 6.1 and related new references have been quoted as new numbers [131,132].

Please note that the new sentences are yellow highlighted and that the reference list has been changed and re-numbered.

Misspelling and mistakes have been corrected throughout the text.